

# Linking individual and population patterns of rocky-shore mussels

Romina Vanessa Barbosa[1], Cédric Bacher[2], Fred Jean[1] and Yoann Thomas[1]

[1] Univ Brest, CNRS, IRD, Ifremer, LEMAR, Plouzané, France
[2] Ifremer, DYNECO, Plouzané, France

## ABSTRACT

Individual traits and population parameters can be used as proxies of processes taking place within a range of scales, thus improving the way we can evaluate species response to environmental variability. In intertidal rocky shores, patterns at the within-site scale, *i.e.,* between centimeters to hundreds of meters, are important for understanding the population response into these highly variable environments. Here, we studied a rocky-shore mussel population at the within-site spatial scale (1) to test how intertidal height and orientation of the shore affect individual traits and population parameters, (2) to infer the link between individual and population level features, and (3) to explore the upscaling mechanisms driving population structure and processes. We analyzed the patterns of six population parameters: density, biomass, crowding, median individual size, recruitment and mortality rate, and four individual traits: growth rate, spawning phenology, size and condition index. Crowding was defined as the degree of overlapping of individuals within a given area, for which we created a "crowding index". Mussels were studied along the intertidal height gradient in two rocky shores with contrasted orientation at one site over a full year. Our results showed a significant effect of intertidal height and shore orientation on most of individual traits and population parameters studied. In contrast, biomass contained in a full covered surface did not vary in space nor in time. This pattern likely results from relatively constant crowding and a trade-off between median individuals' size and density. We hypothesize that growth, mortality and recruitment rates may all play roles in the stability of the crowding structure of mussel aggregations. Variation in spawning phenology between the two shores in the study site was also observed, suggesting different temporal dynamics of microclimate conditions. Interestingly, despite the different population size distribution between the two shores, our estimates indicate similar potential reproductive output. We hypothesize that the structure of the patches would tend to maintain or carry a maximum of biomass due to trade-offs between density and size while maintaining and maximizing the reproductive output. The patterns of spatial variability of individual traits and population parameters in our study site suggest that heterogeneous within-site conditions influence variation in individual performance and population processes. These results provide insights about the relationship between individual traits and how these relationships make patterns at the population level emerge. They provide baseline information necessary to improve models of metapopulation with spatially explicit processes.

Corresponding author
Romina Vanessa Barbosa,
rominavanessa.barbosa@univ-brest.fr

## INTRODUCTION

For marine species living in intertidal zones of rocky shores, environmental conditions vary from centimeters to hundreds of meters depending on habitat structure. These conditions determine population responses through the metabolic processes, behavior, fitness and survival of individuals. For macrotidal ecosystems, the intertidal height creates a gradient of environmental conditions related to the aerial exposure of individuals, which increases from low to high elevation. Consequently, temperature varies according to the intertidal height, with higher daily average maximum at higher intertidal height (*Helmuth et al., 2011*). Temperatures also vary daily, depending on topographic features (*Denny et al., 2011*; *Seabra et al., 2011*; *Dong et al., 2017*; *Miller & Dowd, 2019*; *Wang et al., 2020*), which creates microclimates (*Helmuth et al., 2006*; *Choi et al., 2019*). The mosaic of microclimates on a site can represent higher environmental variability than latitudinal mean temperature comparisons (*Denny et al., 2011*; *Seabra et al., 2015*). Therefore, evaluating individual variability and population response at the within-site scale (*i.e.,* from centimeters to hundreds of meters) could improve comprehension and forecasting of population dynamics at higher scales.

A diversity of species is adapted to living in highly heterogeneous rocky-shore habitats, mussels being among the most important (*Paine, 1971*; *Paine, 1974*; *Menge, 1976*). Mussels are engineer species that create habitat for other species (*Borthagaray & Carranza, 2007*; *Arribas et al., 2014*) with a wide distribution. The strategy used by mussels such as *Mytilus* spp. to dominate the intertidal space is their aggregative behavior (*Bertness & Grosholz, 1985*). Their aggregation structure can be characterized in terms of crowding - how close are individuals - and number of layers (*e.g., Guiñez, Castilla & Sterner, 1999*), features that vary between sites and species, and have been related to environmental conditions and species mobility capacity (*Commito & Rusignuolo, 2000*; *Gutiérrez et al., 2015*; *De Jager et al., 2020*; *Zardi et al., 2021*). Mussel aggregation structure conditions the development and survival of young mussels, by increasing survival of recruits living between aggregated adults (*Bertolini, Montgomery & O'Connor, 2018*). It also conditions the stability of entire patches, since it could affect the resistance to dislodgement (*Denny, 1987*; *Gutiérrez et al., 2015*; *Zardi et al., 2021*). Aggregation structure is, therefore, fundamental for the persistence of the population in diverse conditions.

Features of individual mussels and mussel populations have been more frequently examined at regional or latitudinal scales than at the within-site scale. For instance, mussel density and covered area vary between sites and according to intertidal height distribution as a result of complex mechanisms linked to environmental conditions and biotic interactions (*Blanchette & Gaines, 2007*). Mussel populations also present variation of mean individual size and abundance associated with different growth, likely driven by environmental factors such as food availability, wave exposure and temperature (*Blanchette, Broitman & Gaines*

*2006*; *Blanchette, Helmuth & Gaines, 2007*; *Blanchette & Gaines, 2007*; *Fitzgerald-Dehoog, Browning & Allen, 2012*; *Gomes et al., 2018*). In the intertidal zone of macrotidal areas, the above-described variability of environmental conditions likely determines the performance of individuals across the intertidal height and depending on the angle of orientation of their rocky substrate. Such variability has been observed at latitudinal scales covering a wide area, but scarce information exists about this kind of patterns at within-site scale.

Monitoring environmental conditions and their relationship with individual response at small-scale across a complete study area remains challenging. We propose to use individual traits and populations parameters as proxies of performance to fill the actual knowledge gap and test new hypothesis. Evaluating patterns of individual traits and population parameters at different hierarchical levels and scales (spatial or temporal) is an important way to infer possible underlying mechanisms (*Grimm et al., 2005*). Any observation presenting a non-random structure, such as the observed structure of individual's disposition in the space (*e.g.,* patches, uniform, *etc.*) or the temporal dynamics of individual traits (*e.g.,* phenology of reproduction) constitutes a "pattern". For instance, the aggregation pattern is likely driven by mechanisms of inter-individual interaction and environmental stress (*Van de Koppel et al., 2008*; *De Jager et al., 2011*; *De Jager et al., 2020*).

In this study, we aimed to study a rocky-shore mussel population at the within-site spatial scale. We hypothesized that the intertidal height and the main orientation of the shore create patterns of traits distribution. Given the hydrodynamics and mixing of the water column at within-site scale, we assumed that all other environmental variables such as food or salinity were spatially homogeneous. We analyzed the patterns of six population parameters: density, biomass, crowding, median individual size, recruitment rate and mortality rate, and four individual traits: growth rate, spawning phenology, maximum size and condition index. Our objectives were (1) to test how intertidal height and orientation of the shore affect individual's traits and population parameters, (2) to infer the link between individual and population level features and (3) to explore the possible upscaling mechanisms driving population structure and processes.

## MATERIAL AND METHODS

### Study site

The study site, known as Le Petit Minou, is located in the bay of Brest, on the western coast of Brittany, France. Le Petit Minou site constitutes a sheltered macrotidal area, which presents a tidal height range of about 6 m (ca. 1–7 m height). It is made of rocky shores on either side of a beach (Fig. 1). The two shores show contrasted mean orientation, the "East shore" orientated toward the West–Southwest and the "West shore" oriented toward the Southeast (Fig. 1). Mussel patches are sparsely distributed on the West rocky shore whereas they form larger and almost continuous patches on the East rocky shore. Mussel patches consisted of only one layer of individuals in the studied area. Although populations of hybrid individuals are found in Brittany region (*Bierne et al., 2003*), Le Petit Minou site has been characterized by a dominance of *Mytilus galloprovincialis* (*Simon et al., 2020*).

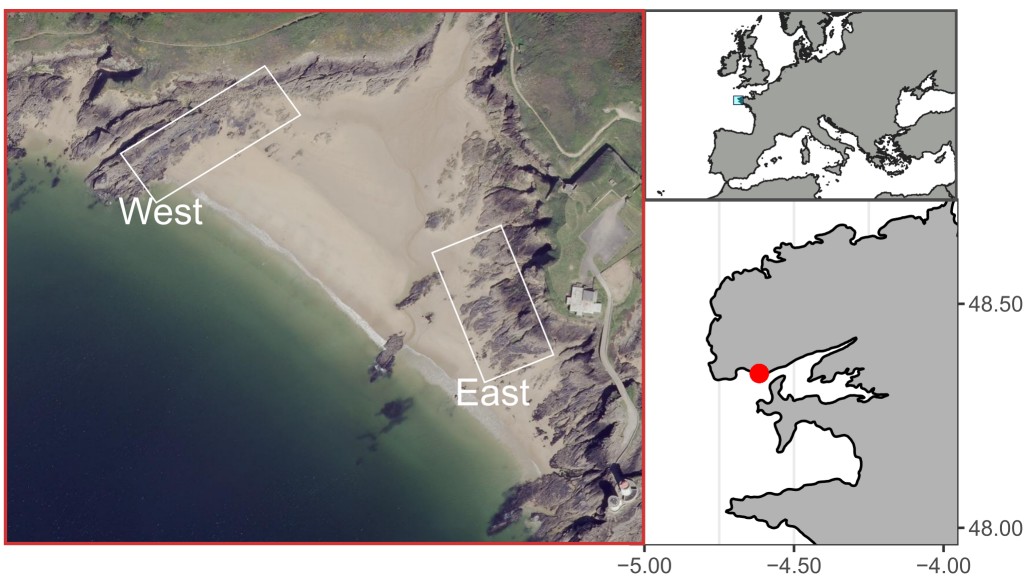

**Figure 1** **Study site at Le Petit Minou, Plouzane, France.** The "West" and "East" correspond to the rocky shore sampling areas to the left and right sides of the beach, respectively. Figure source credit: Barbosa Romina Vanessa, Bacher Cedric, Jean Frederic, Thomas Yoann (2021). Individual traits and population parameters of a rocky-shore mussel population. SEANOE. https://doi.org/10.17882/80337 (CC-BY-NC).

## Mussel sampling strategy

Mussel sampling was carried out on the two rocky shores to either side of Le Petit Minou beach, hereafter referred to as the East and West shores. Sampling was performed monthly from March 2019 to February 2020 by sampling three quadrats (25 × 25 cm) in each shore. The quadrats were positioned to represent different height positions, *i.e.,* a total of 3 quadrats to represent the entire intertidal height gradient, on each shore (West and East). We sampled areas with at least 20–25% of mussel cover in low, mean and high height positions along the altitude range of mussel distribution of each shore (about 1.5 m in the study site). Quadrat location coordinates were recorded using a real-time kinematic (RTK) GPS, with Lambert93 projection. Before sampling, a picture of each quadrat was taken, which was later used to calculate the area of the quadrat covered by mussels by manually delineating the aggregation margins using ImageJ software.

A drone survey was performed with a DJI® Phantom 4 pro in June 2019 to assess the topographic heterogeneity. Drone images were taken at a 17 m altitude with an overlap of 80% between images to optimize the image alignment and produce a relevant digital elevation model (DEM) of the area. We registered multiple reference position points from the mapped area with a RTK GPS to correct for possible distortions in the DEM reconstruction (*Jaud et al., 2018*). The analysis of drone images and GPS position for the photogrammetric process was performed with Agisoft PhotoScan Pro v1.2.3. Mean aspect, a circular land-surface parameter used to obtain a continuous gradient stressing the north-south or east–west gradient (*Olaya, 2009*) was calculated in GRASS GIS software. For

each shore, aspect was represented as "northness" or "eastness" by deriving the topographic feature orientation (a sine or cosine transformation, respectively) from the DEM (*Olaya, 2009*). This approach allowed to extract the quadrat samples height position, northness and eastness at the precise coordinates where quadrats were sampled. Height position was referred to the Lowest Astronomical Tide (LAT; the chart datum used for tides predictions and water height measurements).

## Percentage immersion time

The mean daily percentage immersion time was determined along the intertidal height gradient in the study site based on the tidal regime measurements over the preceding 5 years (from January 1, 2015 to December 31, 2019) in the study area. Hourly tide gauge measurements of water height at Brest (5 km far from Le Petit Minou) were obtained for the given period from the 'Service hydrographique et océanographique de la Marine' (SHOM; https://data.shom.fr/). These measurements were used to fit a cubic polynomial model to compute the mean percentage of immersion time (%) in relation to intertidal height (m). We then determined the immersion percentage of every sample based on its intertidal height position.

## Sample treatment and primary data
### Individual biometric measurements

All mussels in the quadrat samples were sorted using meshes to select the individuals larger than 0.05 cm. Individual shell length (SL, mm) of all selected individuals was then measured by image analysis, using ImageJ software. Pictures of the individuals larger than ca. 0.5 cm were taken at 50 cm height using a canon EOS600D camera, whereas those of small individuals (length < ca. 0.5 cm) were performed under a stereo microscope (ZEISS Stemi 508) with a x6.4 magnification.

For each shore (East and West), thirty individuals were subsampled by selecting ten individuals per quadrat representing its whole range of length (>0.5 cm length), each month for biometric measurements: total wet weight (TWW), flesh wet weight (FWW), flesh dry weight (FDW) and shell length (SL). These individuals were measured using a Vernier caliper and weighed. Their dry mass was then obtained after drying at 60 °C for 48 h (until constant weight). Individual condition index was calculated as $CI = FDW/SL^3$ (*Petersen et al., 2004*).

### Spawning, sex ratio and recruitment

Histological analyses were performed on 10 individuals above 12 mm in length of each shore and date. This minimum length corresponds to the reproductive maturity according to *Toro, Thompson & Innes (2002)* and *Van Haren & Kooijman (1993)*. Their sex and reproductive stage were determined to characterize the spawning period and sex ratio of the population. The individuals were dissected, and the flesh fixed immediately in Davidson's solution. After 24–48 h, the individuals were transferred to 70% ethanol, where they were stored until their inclusion in paraffin and histological preparation. The paraffin inclusion process was performed with a Shandon Citadel 2000 Tissue Processor and paraffin blocks were prepared with a cross-section of each individual. The blocks were then sectioned in a

microtome to obtain 5-μm layers, which were tinted with hematoxylin and eosin. Pictures of the tissues were taken for further individual reproductive stage determination under an inverted microscope (microscope Leica DMIRB) at x25 magnification. To assess the spawning phenology, we determined the spawning/non-spawning stage following *Lubet (1959)*. Results were presented as the proportion of individuals at the spawning stage and non-spawning stages for each sampling date ($n = 10$).

Recruitment was evaluated by identifying the individuals with shell length smaller than 1.07 mm in every sample. These correspond to the length attained at 1 month based on a von Bertalanffy growth function (*Von Bertalanffy, 1938*) calculated from parameters estimated from the site data as a whole (see details in the following section). Recruitment was represented as the monthly recruit density in a fully covered area, *i.e.,* dividing the number of recruits by the area of the quadrat covered by mussels (estimated from the image of the quadrat).

### Individual growth

Individual growth profile was estimated from the seasonally oscillating von Bertalanffy growth function (soVBGF) parameters (*Somers, 1988*). The individual length distribution of the population, considering all sampled individuals, was used to estimate the parameters of the soVBGF based on length frequency analysis (ELEFAN, *Pauly & David, 1980*):

$$L_t = L_{inf}(1 - e^{(-K(t-t_0)+S_t-S_{t0})})$$

where $L_t$ is length at age $t$, $L_{inf}$ is the asymptotic length, $K$ is the von Bertalanffy growth constant, $t_0$ represents the theoretical age when individual length is 0, and $S_t = \frac{CK}{2\pi}\sin(2\pi(t-t_s))$. $C$ is a constant representing the amplitude of the seasonal growth oscillation, with higher values indicating higher seasonal contrast/oscillation, *e.g.,* $C = 0.5$ indicates that growth during the favorable season is increased by 50% (maximum); $t_s$ is a fraction of the year where the sine wave oscillation begins (relative to $t = 0$), *i.e.,* when it turns positive and shifts towards a season when growth is strongest. In the ELEFAN model, length is related to time rather than age, and the time when length is 0, $t_0$, can be interpreted as the recruitment time. $t_0$ takes values from 0 to 1, so 0.5 means a recruitment period in June. Parameter estimation is based on the shell length mode distribution of the population in each shore to account for different distribution shape, in the East and West shores.

The Genetic Algorithm (GA) was applied for the optimization of the fitting process and was implemented with the ELEFAN_GA function from the *TropFish* package in R software (*Taylor & Mildenberger, 2017*). The ELEFAN_GA function was applied with a bootstrap approach with 100 resamplings. A first exploration of the best fitting scores of $L_{inf}$ and $K$ parameter combinations was performed by response surface analysis (RSA) on a wide range of $L_{inf}$ (40–140 mm) and $K$ (0.01–1) values. The range of $L_{inf}$ searching values of the final analysis was restricted then according to the first exploration. In the parameter estimation analysis, one important factor is the moving average (MA) setting, which can affect parameter estimation since it determines the number of bins used for the moving average (*Taylor & Mildenberger, 2017*). A MA of 9 was selected, which corresponds to an approximation of the number of bins spanning the smallest (*i.e.,* youngest) cohort width,

as suggested by *Taylor & Mildenberger (2017)*. This MA selection approach was supported by a sensitivity analysis for evaluation of the results from MA values of 3, 5, 7, 9 and 11.

The consistency of $L_{inf}$ and $K$ parameter results from the ELEFAN_GA approach was validated by comparing simulated growth curves with a size-age relationship of a set of individuals based on sclerochronology analysis. For the sclerochronology analyses, the shells of individuals of diverse size (17 individuals from the East shore and 17 from the West) were selected and fixed in resin. A longitudinal cut was performed on the resin block and the resulting shell slice was observed under a microscope to count the annual growth marks to estimate individual age. This gave one age-length point from each individual in the validation step.

We then compared the growth curves between the West and East shores based on the ELEFAN_GA approach since this analysis considered all sampled individuals. We calculated the overall growth performance phi prime ($\varphi'$) from the estimated $K$ and $L_{inf}$ (*Munro & Pauly, 1983*) for the West and East shores separately, following the equation:

$$\varphi' = \log_{10}(K) + 2\log_{10}(L_{inf})$$

### Mortality rate

The age of individuals on the West and East shores was estimated based on their length and the von Bertalanffy growth curve to obtain the age class distribution. From this, the average mortality rate was estimated for each sample as in *Blicher, Sejr & Høgslund (2013)*, by applying a negative exponential mortality model to the population age distribution:

$$N_t = N_1 e^{-Zt}$$

were $N_1$ is the number of mussels (ind m$^{-2}$) in age class 1, $N_t$ is the number at age $t$, and $Z$ the mortality rate. To avoid bias due to the variability in recruitment density, age class 1 was excluded.

### Biomass, density, crowding and potential fecundity

It is worth mentioning that, in this study, we did not evaluate the covered area as a parameter but we evaluated the structure of the covered area inside the quadrat with diverse population parameters such as biomass, density, crowding, and others derived from biometric traits of individuals (*e.g.*, median shell, see next section). It means these parameters do not represent the real densities or biomass of the entire site but standardized values representing the aggregate composition.

The total biomass of each individual was estimated based on the allometric relationship between SL and TWW, TWW $= a$ SL$^b$. Parameters $a$ and $b$ were estimated by a nonlinear least square function approach applied to the ensemble of biometric observations (n individuals $= 720$) (R software, *nls* function). Then, total biomass per square meter was obtained from the sum of individual TWW (kg) and the percentage of covered area of each sample, determined from the image of the quadrat:

$$\text{Total biomass (kg m}^{-2}) = \frac{\sum \text{TWW}}{\text{CC QA}}$$

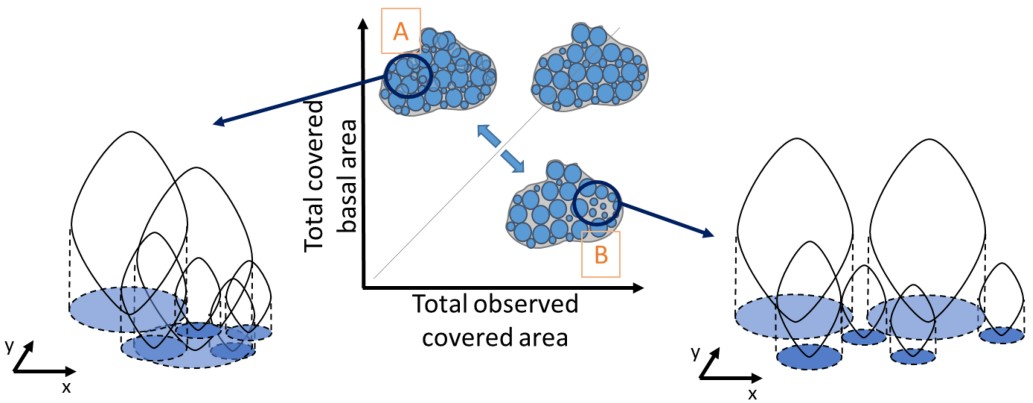

**Figure 2 Schematic representation of the degree of crowding in a patch of individuals as the ratio between the observed covered area and the total basal covered area.** The basal covered area corresponds to the projected vertical shadow of the individuals, represented in blue. A perfect equivalence between these areas (grey line) corresponds to a crowding index value of one, when there is no overlap between basal areas. Very crowded patches have a high overlap of the individual basal areas (A), whereas in less crowded patches there is more space between individual basal areas (B). The crowding index takes values > 1 in case A and < 1 in case B.

where TWW is the total mass of each mussel in the quadrat, CC is the 'proportion of covered area', *i.e.,* the proportion of the quadrat covered by mussels, and QA is the quadrat area $= 0.0625$ m$^2$.

Similarly, density was obtained by dividing the total number of individuals in a quadrat by the mussel-covered area. To examine variations in mussel crowding along the intertidal height gradient, between shores and its relationship with population density and median individual length, a novel "crowding index" was derived. It was defined as the ratio between the observed covered area, corresponding to the visually observed area from the picture analyses (Fig. 2), and the cumulative sum of the basal area of all individuals, derived from the biometric relationship: individual basal area $= \frac{L}{1.8}\frac{L}{2.4}$ (*Alunno-Bruscia, Bourget & Fréchette, 2001*), without including recruits. This makes it possible to represent the spatial crowding of individuals inside a patch. A ratio equal to one indicates perfect equivalence between the covered basal area and the observed covered area. Values higher than 1 indicate patches with a high degree of crowding among individuals (Fig. 2A), and values lower than 1 correspond to lower crowding and consequently greater space between individuals (Fig. 2B). Our crowding index quantifies the overlapping degree between individuals of a monolayer aggregation occurring in the study site and is original compared to other approaches (*e.g., Guiñez, Castilla & Sterner, 1999*).

Finally, the potential population reproductive output was estimated as the quantity of eggs released per square meter considering:

- 1:1 sex ratio,
- the mean mussel size distribution on each rocky-shore,
- the contribution of mature individuals (shell length > 12 mm, *Van Haren & Kooijman, 1993*; *Toro, Thompson & Innes, 2002*),

- a fecundity of 28% dry mass (*Van der Veer, Cardoso & Van der Meer, 2006*),
- and the egg quantity - mass equivalence: $10^6$ eggs = 52.5 mg (*Bayne, Gabbott & Widdows, 1975*).

## Statistical analysis

To identify dates where the frequency of spawning individuals differed between the shores, a G-test of independence was performed between West and East rocky shore areas for each sampling date. A Chi-squared test was performed to test whether the population had a 1:1 sex ratio over the entire study site.

Statistical analyses were performed considering the independent variables: (i) West and East shores (*i.e.,* between-shore differences), (ii) date (*i.e.,* temporal variation), and (iii) intertidal height (in meters, continuous variable). We evaluated the effects of these variables on each population parameter and individual trait (listed in the introduction) by applying general linear models (LM) and a generalized linear model (GLM) in the case of condition index. To fulfil the assumptions of homogeneity of variance and normal distribution of residuals for linear models, a log(base10) transformation was applied to the density of individuals. The homogeneity of variance of groups was verified using the Levene test with *leveneTest* function (*car* package; *Fox & Weisberg, 2019*) and visual observation of the residuals versus fitted values. The normal distribution of residuals was verified by their quantile distribution and a Shapiro test of normality using *shapiro.test* function (*stats* package; *R Core Team, 2020*).

A GLM was performed to test the condition index (CI) using the *glm* function of the *stats* package (*R Core Team, 2020*), with a gamma family and the link function: log, which allows fitting count data with high dispersion. The *lsmeans* function (*lsmeans* package; *Lenth, 2016*) was used to perform all pairwise comparisons using a post-hoc Tukey tests with an adjusted *p*-value and alpha of 0.05.

The spatial variation of the instantaneous mortality rate was tested both along the intertidal height gradient and between the West and East shores, as was the interaction between these factors. This was performed by applying a general linear mixed model (LMM) with log10 transformed response variable and sampling date as a random factor using the *lmer* function (*lme4* package; *Bates et al., 2015*). The effect of adult density on the recruit density was also tested in addition to the effect of shore, date and intertidal height. One LM was done with all these independent variables and a second with the significant factors, adult density and date to identify the months with significant differences.

Finally, the significance of the relationships between crowding index and the population parameters density, median individual length and shore (East/West), were analyzed using a LMM. Density of individuals and median individual length were considered as fixed factors and date as a random factor. The assumption of the normal distribution of residuals were verified with a Shapiro–Wilk normality test. For this analysis, density and median length of population size distribution were calculated without including recruits because the latter do not contribute greatly to crowding.

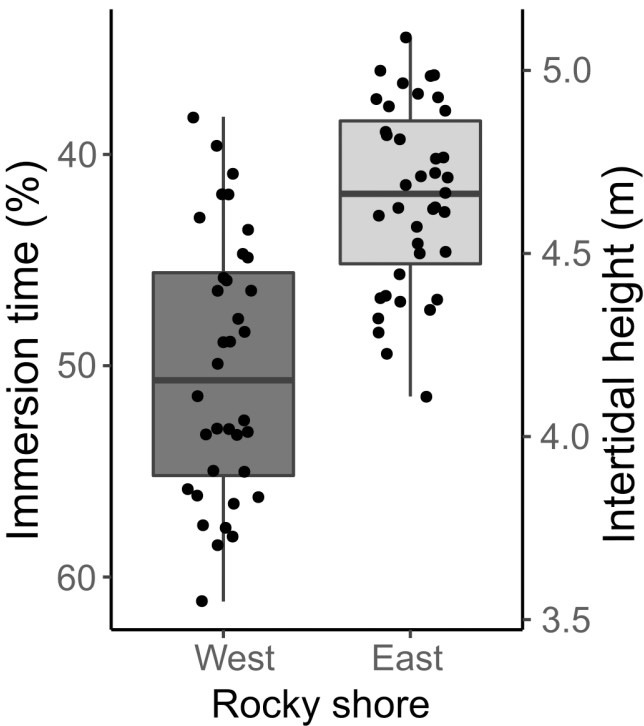

**Figure 3   Percentage immersion time (%; left *y* axis) and intertidal height (m above LAT; right *y* axis) of quadrat samples on the West and East shores of Le Petit Minou.**   The lower and upper hinges of the boxplots correspond to the first and third quantiles, the middle line corresponds to the mean value, and whiskers (vertical lines) indicate the highest and smallest values (between 1.5 * the interquartile range (IQR)). Each point corresponds to one quadrat sample.

## RESULTS

### Intertidal height range and immersion percentage of mussel samples

The samples covered the entire intertidal height gradient on both West and East shores, ranging from 3.55 to 5.09 m, corresponding to a yearly-averaged immersion time varying from 34.47 to 61.16% (Fig. 3). The West shore showed mussel patches along almost the whole of this intertidal height range, whereas the East shore reached the lower limit of mussel distribution at a higher intertidal height (4 m). The orientation of samples on the two shores presented significant differences in eastness and northness features. The samples from the West shore were mainly orientated toward the south and almost perpendicular to the east, whereas the East shore samples were oriented to the west and almost perpendicular to the north-south axis (Fig. S1).

### Condition index, spawning, sex ratio and recruitment

The sex ratio was not significantly different from 1:1 according to a Chi-squared test. The condition index (CI) was only significantly higher on the West shore than on the East shore in August, and there were some significant differences between dates depending on shore (Tables 1, S1, Fig. 4A). CI was significantly lower during March–April 2019 and February 2020 on the West shore, and during March 2019 and February 2020 on the East shore,

**Table 1  Statistical analyses evaluating the shore, date and intertidal height above LAT effect on variables at individual and population scales.** The statistical analysis used is specified in the "Analysis method" column.

| | Independent variables | | | Interactions | | Analysis method |
|---|---|---|---|---|---|---|
| | Shore | Date | Intertidal height | Date*Shore | Intertidal height*Shore | |
| Biomass | n.s. | n.s. | n.s. | n.s. | * | LM |
| $\log_{10}$ (Density) | $3.66 \times 10^{-8}$* | n.s. | 0.0083* | n.s. | n.s. | LM |
| Max. Length | 0.03888* | n.s. | 0.0081* | n.s. | n.s. | LM |
| Median length (excluding recruits) | 0.0001* | | <0.0001* | | n.s | LMM; random factor: Date |
| Condition index | $1.013 \times 10^{-6}$* | $<2.2 \times 10^{-6}$* | n.s. | 0.01575* | n.s. | GLM (family = gamma, link = log) |
| Crowding index | n.s. | n.s. | n.s. | n.s. | n.s. | LM |
| **Crowding index** | | | | | | |
| | Shore | | Adult density | Median length | | |
| Crowding index | n.s. | | <0.0001* | <0.0001* | | LMM; random factor: Date |
| **Recruitment** | | | | | | |
| | Shore | Date | Adult density | Date*Shore | Adult density*Shore | Intertidal height |
| Recruit density | n.s. | 0.0007* | 0.0001* | n.s. | n.s. | n.s. |

Notes.

*Significance and n.s. indicates non significance.

n.s., non significance.

compared with the period June–September 2019 (Fig. 4A, Table S1). Furthermore, CI was not significantly related to the intertidal height (Tables 1, S2). The temporal variation of CI was consistent with the variation in reproductive stage of individuals, with significantly lower CI during the spawning periods (Fig. 4, Table S2). The proportion of individuals at the spawning stage varied through the year with two main periods of increase during March–May and September–November, with some differences between the West and East shores during the second period (Fig. 4B). Differences between the shores were revealed by G-tests in which there was nonindependence of the frequencies of spawning individuals in July ($G = 4.69$, $p$-value $= 0.03$), November ($G = 5.3$, $p$-value $= 0.02$) and February ($G = 4.07$, $p$-value $= 0.04$).

Recruit density indicated an almost continuous recruitment, with significant differences between dates and the same temporal pattern on both shores (Table 1, Fig. 4C). Recruit density also had a nonsignificant relationship with intertidal height, but a significant positive relationship with the density of adult individuals at both shores considered. The relationship with adult density was non-significantly different between shores, *i.e.*, non-significant interaction (Tables 1, S2, Fig. 5). The highest recruitment occurred from April to September, with a significantly higher density in August relative to most other months. Secondary increases were also seen on the West shore in December 2019 and February 2020, although they were not statistically significant (Fig. 4C). The first recruitment

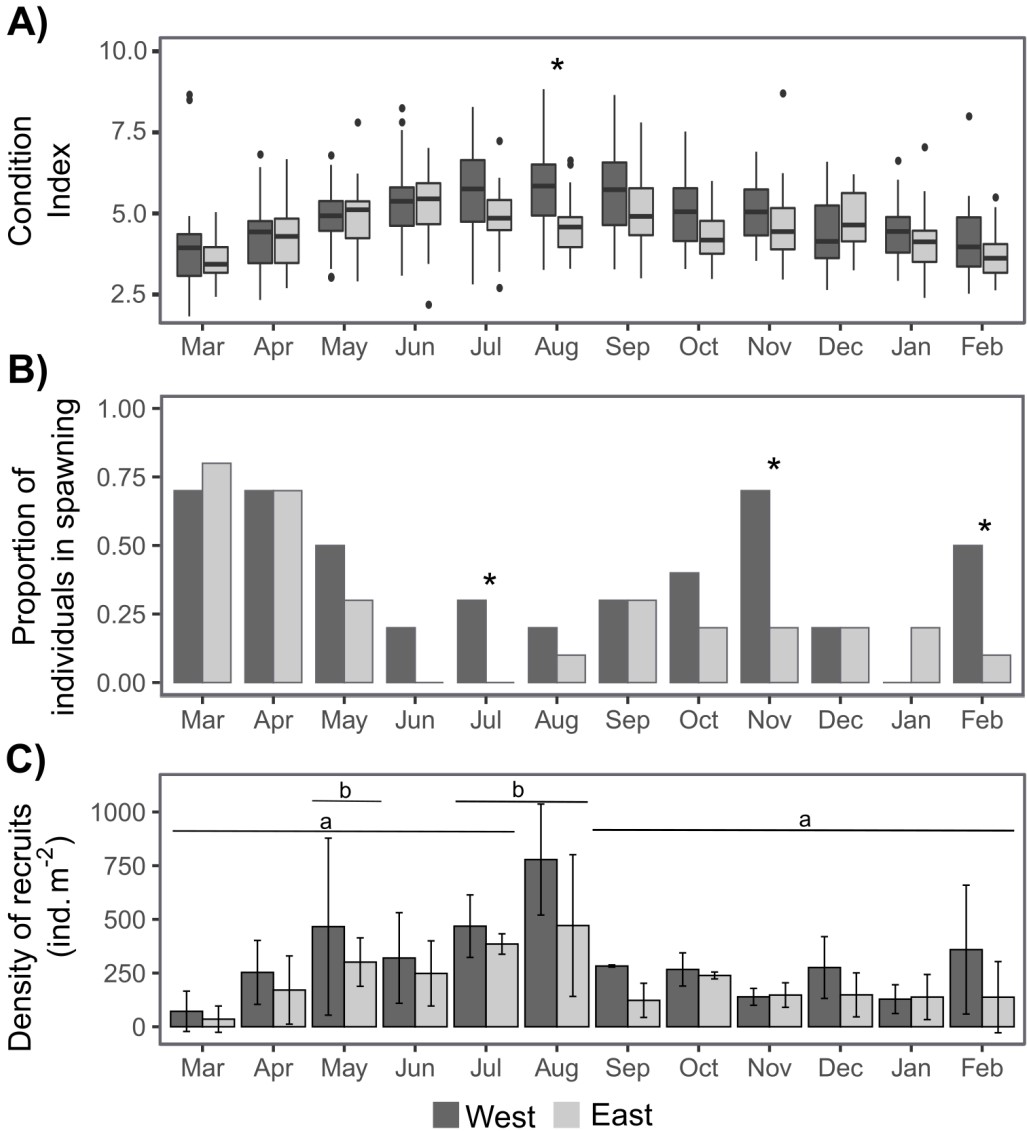

**Figure 4 Annual variation in individual condition index (A), the proportion of individuals at the spawning stage (B), and the recruit density (ind m⁻² month⁻¹) on the West (dark grey) and East (light grey) shores of Le Petit Minou (C).** On (A) the boxplots represent the range between the first and third quantiles and the mean value, with the highest and smallest values (between 1.5 * the interquartile range (IQR)) indicated by the whiskers and individual points represent outliers. On (B) * indicates significant differences between West and East shores ($p < 0.05$). On (C) barplots represent the mean and standard deviation (SD) of each month and shore. Different letters indicate significantly different groups ($p < 0.05$).

periods followed the increase of the proportion of individuals at the spawning stage (March–May 2019), but the higher recruit density was observed 3 months after the end of this period (August). The second increase in the proportion of individuals in spawning stage, September–November 2019, was followed by a small (non-significant) increase in recruit density in December, in the west shore (Fig. 4).

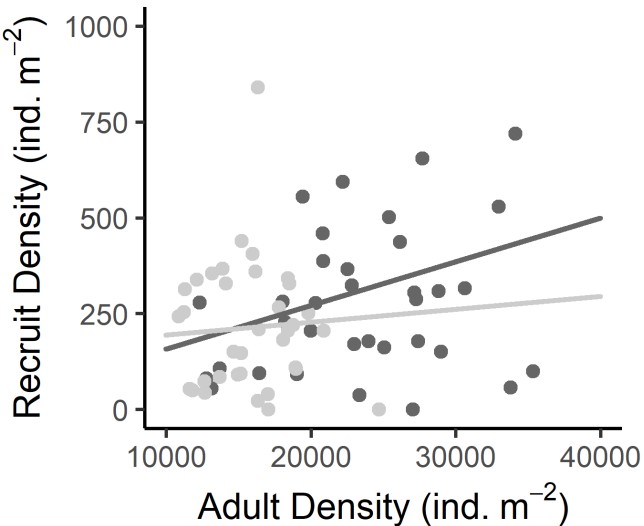

**Figure 5 Relationship between the recruit density and adult density in the West (dark grey) and East (light grey) shores of Le Petit Minou.** Dots represent the data from West (dark grey) and East (light grey) shores at Le Petit Minou. The trend on each shore is represented by the lines; there is no significant difference in the slope of the regression lines (no significant interaction, for more details, see Tables 2 and S3).

**Table 2 Statistical significance of the effects of shore and intertidal height above LAT on mortality rate (Z).** Results according to a linear mixed model (LMM) with date as random factor.

**Mortality rate(LMM)**

|  | Shore | Intertidal height | Intertidal height *Shore |
|---|---|---|---|
| $Log_{10}$ (Mortality rate, Z) | 0.0103[*] | <0.0001[***] | n.s. |

**Notes.**
*Significance and n.s. indicates non significance.
n.s., non significance.

## Individual Growth

The soVBGF parameters indicate differences between the two shores, with $K = 0.12$, $L_{inf} = 65.25$, $C = 0.42$, $t_s = 0.56$ and $\varphi = 2.71$ in the West, and $K = 0.19$, $L_{inf} = 62.54$, $C = 0.51$, $t_s = 0.5$ and $\varphi = 2.87$ in the East (Fig. 6). Both curves are shown in Fig. 6, with the estimated parameters and $t_0$ equal to 0.52 and 0.67 for the West and East, respectively. Moreover, these estimated growth curves matched with the range of the age-at-length data derived from the sclerochronology analysis (Fig. 6).

## Mortality

The instantaneous mortality rate (Z) varied from 0.43 to 1.43 $yr^{-1}$, with a mean of 0.642 (±0.170 SD) in the West and 0.919 (±0.246 SD) in the East. The result from the LMM indicated that the log-transformed mortality rate ($Z$) was significantly different between the two shores and was positively related to the intertidal height in both, *i.e.,* mortality increases as intertidal height increases (Fig. 7, Tables 2, S2).

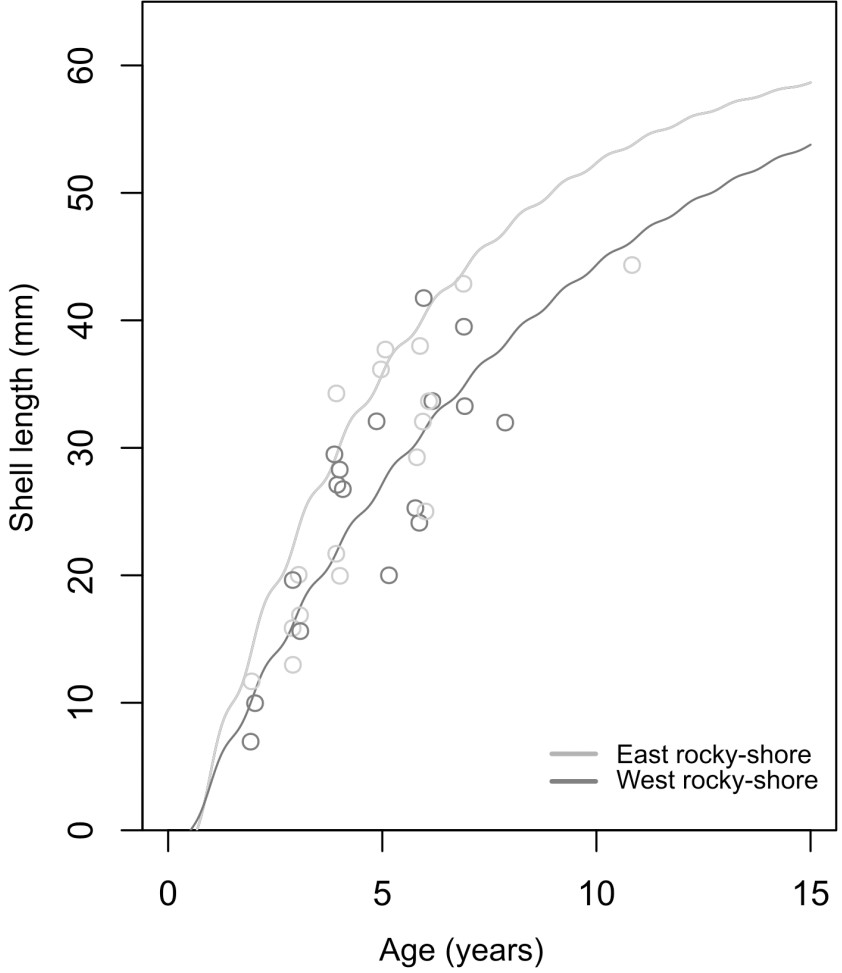

**Figure 6 Estimated von Bertalanffy growth curves observed age-at-length data (circles) showing the concordance between estimations and observations.** The von Bertalanffy curves have been estimated based on the size distribution of all sampled individuals from the West (dark grey) and East (light grey) shores of Le Petit Minou using the ELEFAN_GA approach. Age-at-length data was determined by sclerochronology analysis. Dark grey and light grey circles correspond to the age at length (only the last growth mark was considered) of individuals from the West and East shore ($n = 17$; from each shore), respectively.

## Population biomass, size structure, density and crowding

No significant difference in total biomass was found between the East and West shores or between sampling dates (Tables 1, S2). There was a significant interaction indicating a shore-dependent intertidal height effect on biomass, but this was due to two very influential points in the highest topographic position in the entire site. As these points were outside the common range of distribution between West and East, we did not consider them in the final analysis, so there was no significant intertidal height effect on the biomass (Tables 1, S2, Fig. 8A).

A significantly higher density of individuals was found on the West shore than on the East, without any differences between dates (Tables 1, S2, Fig. 8B). In addition, there was

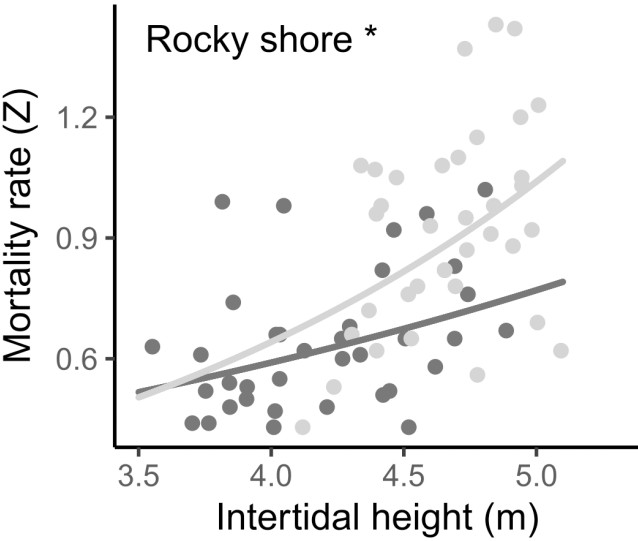

**Figure 7** **Relationship between instantaneous mortality rate (Z) and the intertidal height (m above LAT) of samples on the West (dark grey) and East (light grey) shores of Le Petit Minou.** Lines represent the relationship determined by a general linear model (for more details, see Tables 2 and S3) and dots represent the data from West (dark grey) and East (light grey) shores at Le Petit Minou. Significant difference between shores is indicated: *rocky shore.

a significant intertidal height effect on the density of individuals, with the same positive trend in both rocky-shores (Tables 1, S2–S3, Fig. 8B).

The median length of individuals was significantly higher on the East shore than on the West and was significantly positively related to the intertidal height position on both shores (Tables 1, S2–S3, Fig. 8C). There was a significant higher maximum length in the East than in the West with no differences between dates. Maximum length was significantly negatively related to intertidal height (Tables 1, S2–S3, Fig. 8D).

There were no significant differences in the crowding index either between West and East or between dates (LM results, Table 1, Fig. 8E). Crowding index was positively related to the density of adult individuals and the median length of adult individuals of the population, with a lower slope in the first case (LMM results, Tables 1, S2–S3, Fig. 8F).

### Individual and population fecundity

Considering the mean size distribution of the population on each shore, the potential reproductive output was very similar between the two shores: 112.92 (±42, SD) and 106.5 (±32, SD) g m$^{-2}$ in West and East, respectively (Fig. 9). This was equivalent to a total of $2.15 \times 10^9$ and $2.03 \times 10^9$ eggs m$^{-2}$ released per covered square meter in the West and East, respectively (Fig. 9). The mean total density of reproductively mature individuals (individuals of size > 12 mm, *Toro, Thompson & Innes, 2002*) were 8,819 and 7,939 ind m$^{-2}$ in the West and East, respectively.

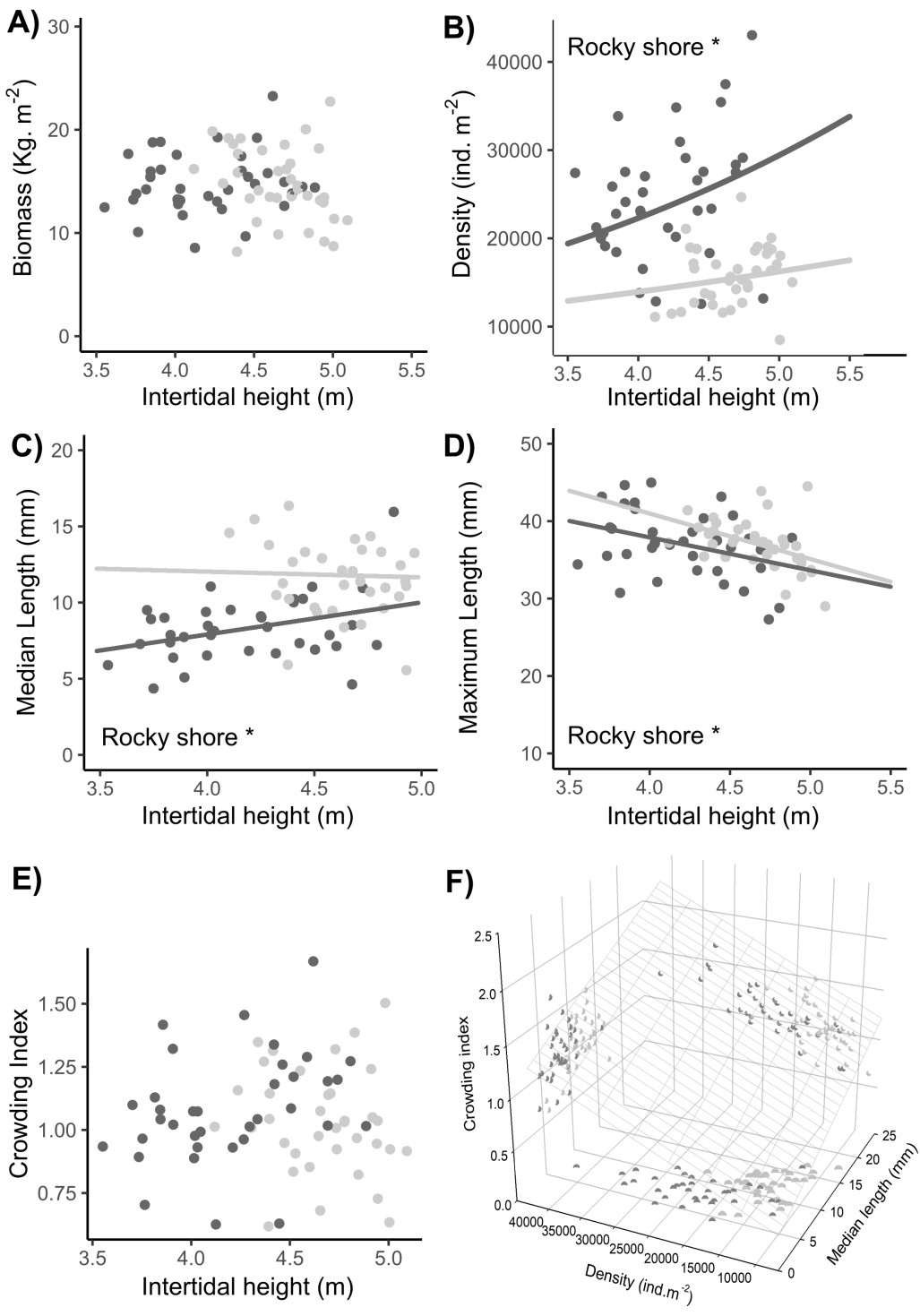

**Figure 8** **Relationships between intertidal height and population variables: (A) biomass, (B) density, (C) median length, (D) maximum length, and (E) crowding index. (F) Relationship of the crowding index with the median length and density of individuals.** Dots represent the data from West (dark grey) and East (light grey) shores at Le Petit Minou; one for each sample (continued on next page...)

# PeerJ

**Figure 8 (…continued)**
($n = 720$). The trends on each shore are represented by the lines in the cases of significant relationships; significant differences between shores are indicated *rocky shore on the plots; there are no significant differences in the slope of the regression lines in each plot (no significant interaction, for more details, see Tables 2 and S3). Median length and maximum length were calculated from a variable number of individuals, which corresponded to the total of individuals in the sample excluding recruits (data available in the supplementary material). In panel (F) the surface layer represents the trend of crowding index depending on the density and median length, and each pair-variables relationship is represented by the projected shadow of observation into each axis (dots).

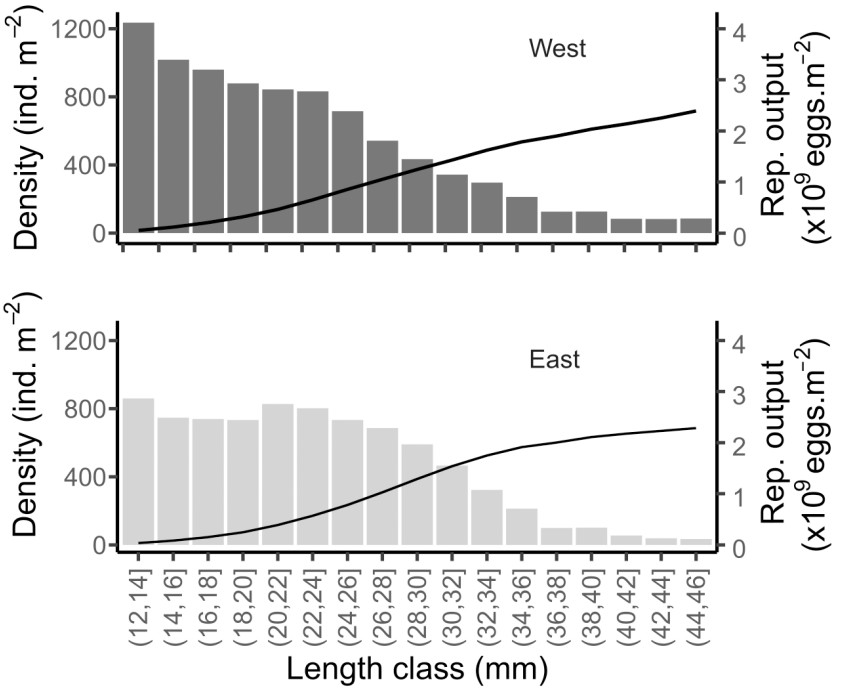

**Figure 9** **Simulated reproductive output of the mussels in the West and East shores of Le Petit Minou.** Data are shown relative to the population size structure (bars) and cumulated (curve), for the reproductively mature individuals (*i.e.,* length > 12 mm).

## DISCUSSION

By combining spatial and temporal observations, we uncovered several patterns related to the phenology of spawning and recruitment, population dynamics and aggregation structure of mussels on two rocky shores. Our results revealed asynchrony between spawning and recruitment, two life history traits of fundamental importance for population persistence. We observed that the spatial variability in spawning phenology allowed a second spawning during higher stress summer conditions. Additionally, our results showed that individual and population patterns contrasted with biomass stability along the intertidal height and between the two rocky shores. In the following paragraphs, we discuss these results and the possible mechanisms underlying the aggregation structure by examining the

relationships among the observed patterns of density, size distribution, growth, mortality and recruitment.

## Phenology of spawning and recruitment

Spawning and recruitment phenology determine the conditions to which larvae and recruits are exposed and will, consequently, affect population dynamics. Recruitment occurs after the spawning period, larvae development and settlement. Larvae may recruit in the same area as their progenitors, *i.e.,* self-recruitment, presumably 2–3 weeks after the progenitors spawned. However, larvae can also disperse and settle in other areas and, in this case, the lack of self-recruitment can be observed as local asynchrony between spawning and recruitment dynamics. Other causes of asynchrony are recruitment failure or a primary recruitment on other sites than the mussel bed, as in filamentous algae (*Bayne, 1964*). Temperature and feeding conditions determine individual spawning and recruitment phenology and success along the coast. In general, there is one main spawning period at high latitudes (*e.g., Kautsky, 1982*; *Antsulevich, Maximovich & Vuorinen, 1999*; *Thorarinsdóttir & Gunnarsson, 2003*) and two periods in temperate regions (*e.g., Seed, 1976*; *Lowe, Moore & Bayne, 1982*). In our study site, spawning, although continuous, peaked in spring and autumn in agreement with prior observations in the Brittany region (*Lubet, 1959*; *Lacroix et al., 2017*). Similarly, recruitment was also continuous and generally low, with peaks in spring and summer. Thus, we observed synchrony between recruitment and spawning dynamics during spring and asynchrony during summer and autumn. During summer, the asynchrony, which was caused by a peak of recruitment without a preceding peak of spawning, could have resulted from the arrival of larvae from other sites with differing reproductive phenology. In autumn, the asynchrony, which involved a peak in spawning without a following peak in recruitment, may be related to the dispersion of larvae to other sites, recruitment failure or recruitment on other substrates not analyzed in this study. These asynchronies between recruitment and spawning highlight the influence of larval dispersal and connectivity in the population dynamics of the study site. *Thomas et al. (2020)* pointed out the fundamental role of connectivity between sites for population dynamics along the coast of this region. Thus, the studied population at this site may have a dynamic, undefined role in overall population connectivity, *i.e.,* acting as a larval supplier or receiver, depending on the season and environmental conditions.

Variation in spawning phenology within the study site was also observed, suggesting different temporal dynamics of microclimate conditions between the two shores. Microclimate conditions may change due to the interactions between factors such as wind speed, wave height, solar radiation, time of low tide and angle of substrate (*Denny, Miller & Harley, 2006*; *Seabra et al., 2011*; *Helmuth et al., 2011*; *Choi et al., 2019*). These factors can be particularly contrasted between West and East shores due to their different orientation. Consequently, individuals showed temporal variations in performance between shores. For instance, both a lack of spawning during July and lower condition index during August suggest a decrease in the performance of individuals on the East shore during summer compared with the West. Summer conditions were characterized by a high maximum aerial temperature (maxima of 27 °C and 25 °C in July and August, respectively,

https://www.historique-meteo.net/france/bretagne/brest/2019/07/), high SST (almost 3 °C higher than the mean of the last nine years, data not shown) and low tide periods at midday. *Choi et al. (2019)* estimated that higher variation in substrate temperature (ranging from 25.7 °C to 41.4 °C) could be due to co-occurring high aerial temperature and high solar radiation, which is likely what took place in our study site. Gametogenesis can be slowed by high temperatures of water (*Fearman & Moltschaniwskyj, 2010*) and seawater temperatures higher than 20 °C can accelerate gamete release (*Múgica et al., 2015*), which may explain the observed discontinuity of spawning between month, and presence of oocyte atresia (data not shown). Also, a low condition index, *i.e.,* loss of individual weight, which is mainly related either to spawning or physiological stress periods, was observed during August (when no spawning occurred) suggesting physiological stress.

Based on these observations, we hypothesize that the observed shore differences in the reproductive stage and condition of individuals were related to a temporal change in the microclimate, specifically the high temperatures recorded. The temperature also drives the reproductive phenology at within-site scale. Our work suggests the likely role of the different shore orientations within the studied site. That is, the different shore orientations potentially create refuges during stressful conditions. The different microclimates between shores were fundamental during summer because they allowed a second spawning period on the West shore, where in general conditions individuals grow slower. This is an example of how, during fluctuating conditions, individuals living in suboptimal conditions can show a higher performance than individuals living in more optimal conditions, *i.e.,* the "suboptimal can be optimal" (*Martin & Huey, 2008*), which ensures their contribution to the gametes that will make the future cohort.

## Variability of traits contrasts with stability of biomass

Unexpectedly, biomass was stable in both space and time in our study. This pattern of biomass stability results from the interaction between the patterns of individual traits and population parameters and is mediated by a relatively constant crowding of individuals. The aggregation structure itself results from a trade-off between the density and size of individuals. More individuals can occupy a given area if they are smaller. This pattern has already been observed in single-layer mussel aggregations (*Petraitis, 1995*). Population size structure and density are affected by both individual- and population-level processes, such as growth, recruitment and mortality. *Petraitis (1995)* pointed out that these processes can drive the capacity of a mussel bed to keep the covered area constant. Keeping the covered area constant is also associated with keeping the crowding constant to maintain the aggregation structure of a 100% covered area. The aggregation structure can remain constant if individual mussels move and tend to segregate, which has been observed in sandy environments (*Liu et al., 2014*) and may play a fundamental role in the aggregation pattern observed on rocky shores. Nevertheless, movement behavior under stressful wave conditions can increase the probability of dislodgement/mortality, as suggested in a *M. edulis* population by *Schneider et al. (2005)*. We hypothesize, therefore, that growth, mortality and recruitment rates can also play a role in the stability of the crowding structure of an aggregation. This is supported by the observed patterns of individual growth rate,
mortality and recruitment throughout the site, between shores and along the intertidal height gradient. The higher individual growth rate observed on the East shore could compensate for the higher loss of covered area or decrease of crowding due to mortality, forming mussel aggregations of larger individuals with a lower density, as observed. Conversely, on the West shore, vacant space can be filled due to higher recruitment, despite lower growth rates forming highly dense aggregations of small individuals. A similar pattern emerged along the intertidal height gradient. The higher mortality observed at increasing intertidal height could be compensated by the higher recruitment, but not likely by the growth rate since individuals grow slower at high intertidal height due to lower proportion of time in immersion. The density-dependent recruitment process observed is fundamental for maintaining the crowding level, particularly under lower growth rate conditions. This is because, when growth is slow, several individuals will fill a determined gap caused by mortality of an individual much faster than a few individuals growing slowly. Density-dependent recruitment has already been observed in other studies (*e.g.,* *McGrorty, Goss-Custard & Clarke, 1993*; *Blanchette, Broitman & Gaines 2006*; *Dolmer & Stenalt, 2010*).

For sessile species on rocky-shores, space is a limiting factor, and the dynamics of space availability are fundamental for understanding population and community dynamics. The crowding structure of the population in our study indicated a highly compacted positioning of individuals. The sum of the basal area of individuals was as much as 50% higher than the observed covered area (crowding of 1.5), which indicates that the general hypothesis that the sum of area occupied by the sum of individuals is equal to the covered area (crowding equal to 1) is not corroborated in wild populations. This assumption was considered for modeling the space occupation in monolayer mussel beds (*e.g.,* *Roughgarden, Iwasa & Baxter, 1985*; *Petraitis, 1995*), and its violation could mean an overestimation of the occupied area and consequently the possible limitation of available space for expansion of a population. Furthermore, the aggregation structure is important for patch stability and population persistence (*Dugatkin, Mesterton-Gibbonsand & Houston, 1992*; *Guichard et al., 2003*; *Liu et al., 2014*; *Guichard, 2017*). For instance, in our study site, the aggregation structure may increase patch stability, as indicated by the longevity of some individuals, found to be around 11 years old, which may underpin the persistence of this population. Thus, representing the mechanisms underlying the spatial occupation and crowding structure of mussel aggregations in rocky environments could allow an exploration of the population response under changing conditions (*e.g.,* *Zardi et al., 2021*).

Interestingly, despite the different population size distribution between the two shores, our estimates indicate similar potential reproductive output (eggs per square meter), suggesting a density-size trade-off. This is to say that high-density patches of small mussels may have a similar egg production as sparser patches of large mussels, under the assumption of similar individual reproductive output (but see *Sukhotin & Flyachinskaya, 2009*) and 1:1 sex ratio observed in the study site. Thus, we hypothesize that the structure of the patches would tend to maintain or carry a maximum of biomass due to trade-offs between density and size while maintaining and maximizing the reproductive output.

### Environmental heterogeneity underlying individual traits and population parameters patterns

The patterns of spatial variability of individual traits and population parameters in our study site suggest that heterogeneous environmental conditions influence variation in individual- and population-level processes. In our site, shore orientation likely determined different temperatures (*e.g., Helmuth, 1998*; *Choi et al., 2019*), whereas intertidal height gradient determine the duration of emersion. During emersion time, orientation heterogeneity and intertidal height could determine the maximum temperature and the duration of exposition to such temperature, respectively. Combining the temperature and duration of exposure is fundamental in regulating the mortality by heat stress (*Harley, 2008*; *Mislan & Wethey, 2015*; *Seuront et al., 2019*). This could be linked to the observed patterns of increasing mortality when increasing intertidal height due to a higher time of exposure. In addition, energetic demand depends on temperature, which could explain the observed differences on growth rate.

Nevertheless, determining the main driver of the observed patterns could be difficult due to the combination of multiple environmental factors. For instance, topographic orientation combined with slope can increase the wave exposition and intensity (*Guichard, Bourget & Robert, 2001*; *Denny et al., 2004*), causing higher mortality by dislodgement (*McQuaid & Lindsay, 2000*) and determine the swash and potential differences in immersion/emersion time. Furthermore, individuals exposed to similar thermal stress conditions (same temperature and duration) show higher survival if they have more food available, *i.e.,* more energy to support the stress (*Fitzgerald-Dehoog, Browning & Allen, 2012*). This reinforces our interpretation of the gradients observed along the intertidal height gradient. Individuals located at low height, with longer immersion times, have greater access to food and consequently lower mortality rates.

## CONCLUSIONS

This study constitutes an integrative characterization of a wild rocky-shore mussel population at both spatial and temporal scales, and at individual to population level. This integrative analysis allowed us to evaluate the patterns and relationships between the main individual traits and population parameters. Moreover, our results shed light on how the interaction of processes at the individual scale could drive the patterns observed at population scale. The next steps needed to gain insight into the system dynamics should address the environmental drivers underlying the variability of processes creating the observed patterns. Analysis of current aerial images to represent and characterize the terrain surface, and to assess the target species distribution (*e.g., Gomes et al., 2018*) may be a suitable approach for identifying the environmental changes at desired scales (microclimates) (*e.g., Choi et al., 2019*; *Kearney et al., 2020*). Finally, linking all the information together, including the patterns, processes and environmental variability, in mechanistic models may be a promising avenue for testing the hypothesis that emerged here, such as the local effect of temperature on the phenology of reproduction, and the mechanisms of crowding and its role in population resistance to stress and disturbances.

## ACKNOWLEDGEMENTS

We thank Eric Dabas for processing the sclerochronology samples and Nelly Le Goic for advice on histological analyses. We thank Jérôme Ammann, Amicie d'Augustin, Even Tangui, Elodie Bessis, Mariana Ventura, Anaïs Medieu, Jordan Toullec, Joan M. Alfaro-Lucas and Yann Kearjean for their help in the field. We thank Helen McCombie for English editing and valuable comments, and the three anonymous reviewers for their comments, which helped us to improve our manuscript.

### Funding

This study benefited from financial support of the Region Bretagne and Université de Bretagne Occidentale (UBO). This project was also funded by the IROCWA project (ANR-19-CE32-0003). The funders had no role in study design, data collection and analysis, decision to publish, or preparation of the manuscript.

### Grant Disclosures

The following grant information was disclosed by the authors:
The Region Bretagne and Université de Bretagne Occidentale (UBO).
IROCWA project (ANR-19-CE32-0003).

### Competing Interests

The authors declare there are no competing interests.

### Author Contributions

- Romina Vanessa Barbosa conceived and designed the experiments, performed the experiments, analyzed the data, prepared figures and/or tables, authored or reviewed drafts of the paper, and approved the final draft.
- Cédric Bacher and Fred Jean conceived and designed the experiments, authored or reviewed drafts of the paper, and approved the final draft.
- Yoann Thomas conceived and designed the experiments, performed the experiments, authored or reviewed drafts of the paper, and approved the final draft.

### Data Availability

The raw data (individual's shell length (36670 registers); biometric measures (720 individuals); quadrat sample information (ex. date, geographic coordinates, covered area, etc); summary of features (ex. the mean density of individuals)) is available at the SEANOE repository: Barbosa Romina Vanessa, Bacher Cedric, Jean Frederic, Thomas Yoann (2021). Individual traits and population parameters of a rocky-shore mussel population. SEANOE. https://doi.org/10.17882/80337.

## Supplemental Information

Supplemental information for this article can be found online at http://dx.doi.org/10.7717/peerj.12550#supplemental-information.

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
