# Peer review of "Linking individual and population patterns of rocky-shore mussels"

_PeerJ, doi:10.7717/peerj.12550_

## Round 0.1 · original submission · Major Revisions

Dear Dr. Barbosa,

Thank you for your submission to PeerJ.

It is my opinion as the Academic Editor for your article - Linking individual and population patterns of rocky-shore mussels - that it is good and interesting work but requires a number of Major Revisions.

The reviewers suggested some changes and reviewer comments are shown below and on your article 'Overview' screen.

Please address these changes and resubmit. Although not a hard deadline please try to submit your revision within the next 35 days.

With kind regards,
Isabel Sousa Pinto
Academic Editor, PeerJ

Reviewer 1 ·

Basic reporting

This is an interesting paper which addressed an existing knowledge gap of within site variability. As the authors disclaim well in the concluding section however this study fails to address the environmental drivers which are responsible for this. These data and information are worth of publication nonetheless and provide a solid foundation for further necessary work.

I have some minor suggestions on the text, that is overall well presented

L27 which scales
L39 what is relatively constant crowding

L64 some definition or at least reference for engineers is needed otherwise this can be omitted

L79 add a reference

Experimental design

I believe all of the data collection was appropriate for the research hypotheses

Validity of the findings

Well written and discussed

Reviewer 2 ·

Basic reporting

This article (Linking individual and population patterns of rocky-shore mussels) describes an attempt to link individual and population-level processes in rocky-shore mussels, based on various parameters measured over a year in one site. The data and the general goal of linking two levels of organization are interesting. The study, however, is essentially descriptive, and it lacks clear hypotheses or research questions. The Introduction, for example, is mostly a recapitulation of the reasons why rocky-shore studies are powerful. I am missing the knowledge gap, and some statement about the contribution that this study is providing. Below you will find comments and suggestions that can hopefully help you make these missing elements more visible.

Experimental design

The design seems appropriate to examine variation of individual and population level responses across time. As I clarify in my comments below, there are several points that need clarification.

Validity of the findings

The findings, as described in the Results, are valid. The Discussion does a good job at interpreting the results in the context of earlier research in the subject.

Additional comments

L33-34: the distinction between 2 rocky shores is a bit bleary. It seems they are 2 areas/plots within a site. I suggest referring to them as plots. Also, instead of referring to ‘the site’, for which we have no information, you can say ‘a site’ or ‘one site’.
Abstract: the abstract explains poorly the specific goals of the study and the results. You mention things like contrasting patterns, but at the end of the paragraph, it is unclear how these patterns help explain the link between the individual and the population levels. I suggest reworking the Abstract with a clearer emphasis on the research question.
L53-59: It is important to specify precisely the spatial scales you are studying here. You first talk about ‘scales ranging from 1 cm to 1 m’. Then you mention ‘hundreds of meters’ in the same context.
L62-67: This sentence can be simplified. Try avoid repeating the concept of engineer, and clarify what you mean by ‘conditions the development’.
L67-70: Unclear sentence. What is it that ‘varies’? The mussels? Their aggregation structure? And what do you mean by ‘structure’ as used in the previous sentence?
L83-107: These two final paragraphs of the Introduction are key for setting the scene of the study. However, I am missing important elements, which makes it difficult to understand why the work was done. As is, you give the impression that the study was a description of patterns at two scales. To strengthen the paper, I think it is key to specify the knowledge gap. Then, if possible, I would love to see some testable hypotheses. We know already that the rocky shore is a useful system to understand mechanisms behind species responses to biotic and abiotic factors. I think the manuscript could go beyond this, for example, by stating predictions about the strength of association between different individual and population level responses.
L118: Were the tide heights really measured? Or did you work with predictions? Please provide a source for these data.
L120: We need more details about how you determined bathymetric position. Did you use a surveyor equipment to measure this accurately for both rocky shores? Did the height differ between the two shores? Also, how did you link the tide data and the measured shore level? Did you validate it in situ?
L124-126: Its unclear what the different bathymetric positions are. Did you use 3 quadrats to represent the entire vertical gradient? Or three quadrats per shore level?
L127-128: Replace ‘picture of’ by ‘picture’. And what do you mean by ‘manual delimitation’?
L144: The larger animals were 0.5 cm? Seems really small.
L151: Did you really dry them at 160 degC? Rather 60 degC?
L166: Indicate the number of individuals used.
L172: Unsure what you mean by ‘fully covered area’. Why should recruitments on bare rock be excluded?
L200: Replace ‘An MA’ by ‘A MA’.
L205-214: The description of the ELEFAN model is clear; however, it is unclear how you produced the data used to parameterize the model. You had 17 animals of different size per shore. What was the raw data extracted from the animals? Did each animal provide 1 datum per growth ring interval, i.e., lengths vs growth rate? Please clarify.
L231-234: I got confused because you previously say that the data considered the variability in mussel cover across quadrats, but then you state that the estimates correspond to a full covered quadrat. Please clarify.`
L240-238: The terms ‘visually covered’ and ‘covered’ are equivalent. Please propose better descriptors for the two terms. I looked in the reference you provide, but the formulation of ‘crowding’ is different than yours. Please use a suitable reference.
L246-250: Extremely long sentence.
L255-256: The sentence ‘In addition, we tested for differences between shores; temporal variation; and the effect of adult density on the spatial distribution of recruit density’ seems unnecessary, given that you describe those effects bellow.
L266: I’m confused why you use a package for linear mixed-effects models to fit GLMs.
Also, why don’t you cite the references for only some of the packages used?
L285: ‘Immersion percentage’ is a bit misleading. Please specify that you are talking about time.
L209: The word ‘exposition’ does not add to the sentence.
L299-301: Please specify what you mean by significantly lower. Lower than which months? Perhaps simply describing them as the period of lowest values would suffice.
L313-315: The effect of the interaction between adult density and shore on the recruitment density is also interesting to examine. It seems that the slopes are crossing, suggesting that the effect of adult density depends on shore.
L313: Instead of saying ‘regardless of which shore was considered’, I suggest ‘at both shores considered’.
L319: Can you specify the date of this second increase?
L325-327: As mentioned in the M&M, it is not clear if you used the sclerochronology data to derive the von Bertalanffy growth curves. Here it sounds as if the curves emerge from a different dataset, while in Fig. 6 caption it sounds as if they were originated from the schlerochronology data.
L337: Standardized to what? Area?
L346: For every reference of bathymetry, you need to be careful about the direction. Bathymetry implies a depth, not a height. Is this how you treat them here? Is that how you treat it in Figure 3? On the intertidal zone, we usually speak about height above the mean lower water mark. Can you clarify?
L348: Why do you refer to ‘median length of individuals’? I imagine that each point in Fig. 8E corresponds to an individual. Or are those really medians? You also provide the maximal length, so perhaps this is an estimate for several animals. In that case you need to clarify number of individuals used.
L391-392: Do you mean asynchrony between recruitment and spawning dynamics? I suggest you specify it.
L417: Here you refer to air or sea water temperature?
L428-431: You use the term ‘variability’ quite loosely. What do you mean by variability of microclimates, for example? Or, why would more variable performance allow a second spawning? Please revise.
Discussion: I think the Discussion was much clearer than the Introduction. I suggest you bring some of the elements from the Discussion into the Introduction. This could provide the base for testable hypotheses.
Figures: Try using the same color code to distinguish shores across figures.
Fig. 3: One of the points is surrounded by a blue halo. Please indicate the reason for this.

Reviewer 3 ·

Basic reporting

no comment, see below

Experimental design

no comment, see below

Validity of the findings

I do not see a statement of data availability in the manuscript. Presumably the original data will be archived in an appropriate publicly-accessible repository. The original data are not available for review with the manuscript.

Additional comments

I have reviewed the manuscript and find it generally well written and describing a substantial data set. I have some points of clarification that I would like to see addressed, and some corrections made to the text and figures. Below I list three main concerns/suggestions, and then a series of points addressed by line number.

1. I’ll start with a possibly embarrassing question: What species of mussel was being sampled in this study? Based on the location I would guess Mytilus edulis or Mytilus galloprovincialis, but based on searching for the terms “Mytilus” or “edulis” in the document, I can’t actually find an explicit listing of what species was present at the site.

2. For the haphazard quadrat placement (lines 122-128), please provide more description of how the quadrat placement proceeded. A total of 3 quadrats per site per month was being used to characterize the entire shore height gradient at these sites, and presumably there is some variation in mussel bed patchiness at the small scale (centimeters to meters) that affects the various measurements being made. Was there any pre-filtering of potential quadrat placement sites, based on characteristics such as requiring some minimal number of mussels being present (i.e. excluding shore heights above or below the extent of the mussel bed at each site that would be bare of mussels, or excluding patches within the mussel zone that might have no mussels due to other characteristics such as being in a deep crevice)? Did the sampled shore height distribution differ at a site across time?

3. I would appreciate the addition of a supplementary table with the coefficient estimates provided for each of the final LM, GLM, and LMM models. The manuscript currently includes tables of p-values, but not the actual coefficient estimates that are used to represent the various slopes shown in the figures.

Line 57: The Denny et al. 2011 reference would also be appropriate here where you discuss the within-site vs. across-latitude temperature ranges.

Lines 79-82: The discussion here of how environmental conditions might affect differential growth leaves out the potentially very important role that food plays in stress tolerance and growth. This paper makes the argument that shore height (bathymetric gradient) will play a role in mussel success, and that immersion time might be nearly doubled from the lower to upper reaches of the mussel zone (34% - 61%), which must affect time available for feeding, and thus access to food. The authors should discuss the potential interactive effect of food supply and other environmental stressors. A relevant reference on the importance of food ration for tolerating environmental stress would be Fitzgerald-Dehoog, Lindsay, Jeremy Browning, and Bengt J. Allen. "Food and heat stress in the California mussel: evidence for an energetic trade-off between survival and growth." The Biological Bulletin 223.2 (2012): 205-216.

Line 287: Either here, or in the Methods, please list what the local reference datum is for tide heights (is it mean low water?, mean sea level? etc.) and also what the full tidal range is (i.e. 0-6m). This will give the reader some frame of reference for what these shore height measures mean.

Lines 286-288: Based on my reading of the methods, I’m guessing that these estimates of immersion percentage are based only on still water levels, rather than incorporating wave run-up and swash. Please state explicitly whether any allowance for wave action is incorporated in these estimates, or if they are solely based on still water level. Based on the site orientation, I would assume that the East site is more directly exposed to ocean swells and might experience more splash and wave run-up.

Line 294: The line here says that East shore samples were almost perpendicular to the east-west axis. Based on Figure S1, should this line instead be almost perpendicular to the north-south axis?

Line 355: The text states that crowding index is negatively related to median length, but looking at Figure 8E it seems that the plane trends positive with increasing median length. This is an example of a case where having an additional table of the regression coefficient estimates would be helpful in checking what the text says.

Line 463: I would recommend replacing ‘fixed’ with ‘sessile’ here.

Line 489: The word “exondation” here could be translated to “emersion” to indicate the period of time when the mussels are expose to the air and not submerged by water.

Figure 2: The diagram here and the discussion Lines 436-479 seem to represent these beds as single-layer mussel beds, rather than multi-layer beds where top-layer individuals are not directly anchored to the rock substratum, but instead grow anchored on top of other adult mussels. Is that an accurate interpretation of the discussion and description of the mussel beds at your field sites? Are the mussel beds at this site limited to a single layer? Building multi-layer beds allows for an expansion of aggregation size without needing an increase in available substratum space, but the discussion seems to exclude this as an option for the populations at these sites. Mislan and Wethey 2015 present the case where a multilayer bed provides benefits to some mussels for avoiding heat stress. Mislan, K. A. S., and David S. Wethey. "A biophysical basis for patchy mortality during heat waves." Ecology 96.4 (2015): 902-907.

Figure 3: The axis title on the right side should say “altitude” rather than ‘altitud’

Figure 5: The units shown on both axes should be m^-2, not m^2

Figure 8A: biomass axis title should be m^-2, not m^2

Table S1: Correct spelling of CONDITION INDEX

---

## Round 0.2 · Minor Revisions

Dear Authors,

Thank you for submitting the revised text. The reviewers are now happy with the manuscript but would like you to make a few more changes. When they are done the manuscript can be accepted and does not need to return to the reviewers

Reviewer 2 ·

Basic reporting

No comment

Experimental design

No comment

Validity of the findings

No comment

Additional comments

Review PeerJ-60692_v1

The authors of this manuscript have addressed all of my previous comments. I only have a few minor comments listed below for your consideration.

L29: 'exhibited stability pattern' is unclear. I also think that the word 'compensation' in line 30 is ambiguous, as it implies an active response. I don't think, for example, that mortality, which is mentioned in the following sentence, is a compensatory response.
L123: I don't agree that the quadrats were triplicates because they were placed on different shore levels.
L144: I believe is 'water height measurements' instead of 'measures'. Same comment applies elsewhere.
L199-200: Modify to 'where the sine wave oscillation begins'.
L296: Unclear what 'CI' stands for.
L298-300: Why did you only apply the post-hoc test when the interaction term was significant? Where you not interested on single term pairwise comparisons?
L454-455: You suggest that your data showed the presence of atresia, but that is not something you measured.
L474: As mentioned above, the term 'compensation' implies an active response. I suggest something like trade-off.

Reviewer 3 ·

Basic reporting

See below

Experimental design

See below

Validity of the findings

See below

Additional comments

I have reviewed the revised version of this manuscript, and I am satisfied that the authors have addressed my original concerns. The additional statistical results and parameter estimates given in the new supplemental tables are helpful. I have only a few minor comments on wording in the new version.
Line 116-117: This could be phrased more clearly as “Mussel patches consisted of only one layer of individuals in the study area.”
Line 126: Replace “coverture” with “cover”

---

## Round 0.3 · accepted · Accept

Dear Dr. Barbosa

Thank you for revising your manuscript using the minor comments from the reviewers. It is now accepted on our part!

With my best regards
Isabel Sousa Pinto